# Combination of the Glutaminyl Cyclase Inhibitor PQ912 (Varoglutamstat) and the Murine Monoclonal Antibody PBD-C06 (m6) Shows Additive Effects on Brain Aβ Pathology in Transgenic Mice

**DOI:** 10.3390/ijms222111791

**Published:** 2021-10-30

**Authors:** Torsten Hoffmann, Jens-Ulrich Rahfeld, Mathias Schenk, Falk Ponath, Koki Makioka, Birgit Hutter-Paier, Inge Lues, Cynthia A. Lemere, Stephan Schilling

**Affiliations:** 1Vivoryon Therapeutics N.V., Weinbergweg 22, 06120 Halle, Germany; ilues@me.com; 2Fraunhofer Institute for Cell Therapy and Immunology, Department of Drug Design and Target Validation, Weinbergweg 22, 06120 Halle, Germany; Jens-Ulrich.Rahfeld@izi.fraunhofer.de (J.-U.R.); mathias.schenk@izi.fraunhofer.de (M.S.); 3Department of Neurology, Brigham and Women’s Hospital, Harvard Medical School, 60 Fenwood Rd., Boston, MA 02115, USA; falk.ponath@helmholtz-hiri.de (F.P.); makiokakoki@yahoo.co.jp (K.M.); clemere@bwh.harvard.edu (C.A.L.); 4QPS Austria GmbH, Department of Neuropharmacology, Parkring 12, A-8074 Grambach, Austria; Birgit.Hutter-Paier@qps.com; 5Anhalt University of Applied Sciences, Bernburger Straße 55, 06366 Köthen, Germany

**Keywords:** glutaminyl cyclase inhibitor, anti-pyroglutamyl β-amyloid antibody, drug combination, Alzheimer’s disease, hAPPsl×hQC mice, immunotherapy

## Abstract

Compelling evidence suggests that pyroglutamate-modified Aβ (pGlu3-Aβ; AβN3pG) peptides play a pivotal role in the development and progression of Alzheimer’s disease (AD). Approaches targeting pGlu3-Aβ by glutaminyl cyclase (QC) inhibition (Varoglutamstat) or monoclonal antibodies (Donanemab) are currently in clinical development. Here, we aimed at an assessment of combination therapy of Varoglutamstat (PQ912) and a pGlu3-Aβ-specific antibody (m6) in transgenic mice. Whereas the single treatments at subtherapeutic doses show moderate (16–41%) but statistically insignificant reduction of Aβ42 and pGlu-Aβ42 in mice brain, the combination of both treatments resulted in significant reductions of Aβ by 45–65%. Evaluation of these data using the Bliss independence model revealed a combination index of ≈1, which is indicative for an additive effect of the compounds. The data are interpreted in terms of different pathways, in which the two drugs act. While PQ912 prevents the formation of pGlu3-Aβ in different compartments, the antibody is able to clear existing pGlu3-Aβ deposits. The results suggest that combination of the small molecule Varoglutamstat and a pE3Aβ-directed monoclonal antibody may allow a reduction of the individual compound doses while maintaining the therapeutic effect.

## 1. Introduction

Dementia currently affects about 50 million people worldwide, representing the 5th leading cause of death [1]. Among the different causes of dementia, Alzheimer’s disease (AD) is the most prominent. At the molecular level AD is characterized by two main pathological hallmarks in the brain, extracellular amyloid plaques, mainly consisting of different molecular forms of the APP derived amyloid-β peptides, and formation of neurofibrillary tangles, composed of aggregated and hyperphosphorylated tau. According to the widely accepted amyloid hypothesis, formation of toxic Aβ oligomers and plaques precedes the tau pathology and is probably an early driver of the disease.

There are a couple of Aβ-directed monoclonal antibodies in the development of treatment of AD. Moreover, the recent promising results from clinical studies with e.g., Lecanemab (BAN2401), which targets large soluble Aβ protofibrils, or Donanemab, which is directed to pGlu3-Aβ, and the accelerated approval of Aducanumab (a human monoclonal antibody preferentially binds Aβ oligomers and fibrils) provide prominent support for the amyloid hypothesis and give rise to the assumption that targeting Aβ is a valid approach to developing treatments for Alzheimer’s disease [2,3,4,5]. Among these antibody molecules, Donanemab recently showed significant removal of amyloid load and cognitive stabilization in a Phase 2 clinical trial [6,7]. Interestingly, this is also the first antibody in clinical development, which specifically binds pyroglutamate-modified amyloid peptides (pGlu3-Aβ, AβN3pG). In principle, the rationale to target pGlu-Aβ is based on the observation of a particular toxicity of this molecular species of Aβ [8,9,10,11]. Numerous studies have shown that formation of certain N-truncated Aβ is followed by glutaminyl cyclase (QC) catalyzed conversion of glutamic acid into pyroglutamic acid to form pGlu-Aβ [12,13,14,15,16,17,18]. This N-terminal modification induces changes in the molecular properties, likely driven by an increase of hydrophobicity, which, in turn, leads to accelerated formation of highly neurotoxic oligomers [8,19]. The toxic effect is potentially caused by a unique interaction of pGlu-Aβ with receptors [20]. In addition, the pGlu-modified Aβ is stabilized against N-terminal degradation causing accumulation in AD brains [21]. Importantly, evidence was provided that the appearance of pGlu-Aβ is interconnected with early stages of AD development and its accumulation parallels cognitive decline [22,23].

Hence, approaches to target pGlu-Aβ were suggested to interfere with AD progression. In principle, two strategies were discovered and successfully evaluated; reduction of pGlu3-Aβ formation by inhibition of QC, or the application of anti-pGlu3-Aβ antibodies [24,25,26,27,28]. In this regard, QC shows several characteristics that link its activity to AD development and progression. Under physiological conditions, the enzyme shows its highest expression in the brain [29,30]. Its expression has been shown to be significantly increased in AD, especially within the cortex [12,31]. The increase of activity parallels with the detection of pGlu-Aβ [12,24]. Accordingly, inhibition of QC has been shown to attenuate AD-like symptoms in mice dose-dependently [27] and Varoglutamstat, a first in-class QC inhibitor, has been proven safe and showed signs of efficacy in clinical phase 1 and 2 studies [32,33]. In contrast to the suppression of pGlu-formation by inhibition of QC, the targeting with antibodies aims at clearance of pGlu-Aβ after formation and/or the blocking of aggregation [34]. Hence, both treatment paradigms address the same goal by apparently different and independent molecular strategies. Therefore, we here aimed at a combination treatment to address, whether the effect of both compounds can be combined.

The result of our study in an AD mouse model overexpressing human amyloid precursor protein containing the Swedish and London mutation and human QC (hAPPsl×hQC) indicates an additive effect of both treatment strategies which might open up new avenues for an AD treatment by combining two disease-modifying Aβ-directed therapies in a practical clinical setting in future.

## 2. Results

The aim of this study was to evaluate the combination effect of a glutaminyl cyclase inhibitor and a pGlu-specific antibody on the formation and clearance of pGlu-Aβ in transgenic mice. Based on the results of previous studies [27], we selected a dose of 0.8 g/kg PQ912 in chow, which translates in this study to a daily dose of ≈140 mg/kg. The pGlu-specific antibody m6 (murine PBD-C06, IgG2a) was also investigated in detail in previous studies [34,35], however, these studies were not performed in the hAPPsl×hQC transgenic mice, which harbor an increased pGlu-Aβ formation. Hence, we first performed a dose-finding study with the m6 antibody to be able to select subtherapeutic doses of both treatments for the combination experiment.

### 2.1. Dose Finding Study with m6 Antibody

In this dose-finding study 150 and 500 µg of the pGlu3-Aβ specific m6 antibody were applied weekly i.p. for 16 weeks. The m6 treatment showed a dose-dependent significant decrease of Aβ42 (*p* < 0.05 for both doses) and pGlu3-Aβ42 (*p* < 0.01 for 500 µg dose) levels in the soluble [Tris buffered solution (TBS)] brain fraction. Lower levels of both Aβ forms were found in the insoluble [sodium dodecyl sulfate (SDS) + formic acid (FA)] brain fraction, too, but this decrease did not reach significance (Appendix A). From these results, we concluded that the lower m6 dose (150 µg/week) results in minimal efficacy (decrease in Aβ), which may be improved with the combination with a QC inhibitor.

### 2.2. Combination Treatment of PQ912 and m6 Antibody

#### 2.2.1. Effect on Animal Health and Drug Levels

In the combination study over 16 weeks all treatments including the combination were well tolerated. Body weight increase over treatment period indicated no statistically significant difference between groups (not shown). Brain levels of PQ912 and m6 antibody were determined at study end to check if these are comparable between the respective treatment groups (Appendix A). The mean brain concentrations of the m6 antibody were all in the same range and independent of the treatment or gender. Although, 2-way ANOVA indicates that gender influences the levels of PQ912 (lower levels in males, *p* = 0.004) there are no differences in PQ912 level between the mono- and the combination therapy (*p* = 0.76).

#### 2.2.2. Development of Aβ Pathology in 9- to 12.5-Month-Old APPslxhQC Mice

Initially, we investigated the development of pathology (Aβ deposition in brain) from baseline to the end of the 16-week treatment period and analyzed if the Aβ levels in the isotype antibody treated control are comparable to that of the vehicle-treated control. Furthermore, sex was evaluated as a possible factor influencing pathology (Appendix A). Here, male mice showed much lower Aβ deposition compared with female animals. For females, a significant increase of Aβ42 (*p* ≤ 0.002) and pGlu3-Aβ42 (*p* < 0.001) during the 16-week treatment period could be observed for the vehicle as well as for the isotype-treated group without a detectable difference between vehicle and isotype antibody treatment. In the male animals also a nominal increase of insoluble Aβ was observed in both groups compared to the baseline but this was less than half of that observed in females, and the differences to baseline were not significant. The low *p*-values of the interaction terms (*p* = 0.1 and 0.03 for Aβ42 and pGlu3-Aβ42, respectively) also indicate that the development of pathology was different between males and females in this experiment. Such a gender difference of Aβ deposition rate was also described for other AD models e.g., APP23 mice [36]. Based on these results, we decided to use the isotype-treated group as adequate control group and only the female animals for detailed analysis of the combination effect.

#### 2.2.3. Effect of Single and Combination Treatment on Aβ Deposition

The different treatments were compared with the isotype control (Figure 1). For both soluble and insoluble brain fractions, a nominal decrease of Aβ42 as well as pGlu3-Aβ42 could be observed for the single treatments. In each case, the decrease was stronger for the combination treatment, and it becomes significant for both Aβ forms in the soluble fraction (*p* ≤ 0.01) and for pGlu3-Aβ42 also in the insoluble fraction (*p* = 0.02).

The results of the biochemical analyses are corroborated by the results of the Aβ histochemistry which are shown in Figure 2. Slices of right brain hemispheres of the female animals were stained for pGlu3-Aβ (K17) and Aβ(1–x) (82E1) (see Appendix A for representative images). The Aβ stainings verify the significant increase of pathology over the 16-week treatment period (*p*-values < 0.01, *t*-test baseline vs. isotype control, Figure 2). In general, lower mean Aβ staining was observed for the treatment with the single agents (PQ912, m6) or the combination. The strongest decrease was observed for the pGlu3-Aβ staining with a significant effect for the combination treatment (*p* = 0.025).

### 2.3. Evaluation of Combination Effect According to Bliss

The Bliss combination indices (CI_Bliss_) were calculated as described above for Aβ fractions where a robust reduction (significant ANOVA) of Aβ was observed. Table 1 shows the effects (% response in comparison to the isotype control) of the single and the combination treatments and the response expected under the Bliss independent conditions (Bliss additivity) as well as the combination index which is calculated by dividing the expected (Bliss additivity) by the observed effect of the combination. Under the assumption that a complete inhibition of pGlu-Aβ formation is possible, combination indices between 0.87 and 0.99 were calculated. The indices support an additive (CI_Bliss_ = 1) to slightly synergistic effect of the combination compared with the single treatments.

## 3. Discussion

Compelling evidence suggests that passive immunotherapy represents a promising approach to treat Alzheimer’s disease [4,6,38,39,40,41]. Among the antibodies which are in advanced clinical development or have recently been approved within the US, it is interesting to note that all of them are specific for aggregated and/or modified forms of Aβ. Donanemab is the first antibody, which binds to a highly toxic post-translationally modified Aβ, pGlu3-Aβ [28]. The peptide has been shown in numerous studies to significantly accumulate in Alzheimer brain [42,43,44]. The N-terminal truncation and modification renders the peptide more hydrophobic and prone to aggregation [11,19]. Importantly, some reports imply that the strong formation of pGlu-Aβ appears at stages of pathology closely before manifestation of clinical symptoms, and accumulation correlates with disease better than the “plain” Aβ plaque pathology or accumulation of full-length Aβ [12,22]. These findings initiated the development of strategies targeting pGlu-Aβ by small molecule QC inhibitors or specific antibodies [24,25]. Both treatments have been assessed intensively in mouse models, providing evidence for a reduction of pGlu-Aβ and concomitant functional improvement in spatial learning [24,27,35].

Although the development of combination therapies for Alzheimer’s disease is highly recommended, [45,46] until now only combinations of approved symptomatic therapies with memantine and acetylcholinesterase inhibitors have been investigated in detail [46,47,48,49,50]. Investigations of combinations with disease-modifying drugs are rare. Similar to our approach, Strömberg et al. [51] tested a combination of two Aβ targeting approaches, a BACE inhibitor and a γ-secretase modulator, and found an additive effect on reduction of Aβ formation in an acute mouse model. However, several aspects support a combination treatment in AD. One prominent supporting feature might be drug safety. For example, with an additive effect of the combination, half dose of each component would achieve a similar degree of therapeutic effect but would keep the individual components more certainly within their respective safety dose range (“therapeutic window”). Especially with regard to ARIA-E (amyloid-related imaging abnormalities—edema) observed with monoclonal antibodies, this feature might be important. Second, tackling pathological Aβ variants or forms by a double-pronged approach could enhance (maximally) the achievable effect by different means. First, interfering very proximal of the pathological pathway by inhibiting production of pGlu-Aβ and second, in combination with a more distal effect reducing already existing pGlu-Aβ oligomers or -plaques. This could more markedly slow the progression of the disease over a broader dynamic progression range.

There are different methods to assess the effects of drug combinations and interpretation of results depend on the respective definition of additivity. Foucquier and Guedj [52] provided a concise summary of the respective methodological landscape. Beside dose-effect-based strategies based on Loewe additivity [37,53,54,55] which need an accurate estimation of dose-effect curves and therefore substantial demand on experimental conditions and number of animals, some simpler effect-based approaches are described (highest single agent, combination sub-thresholding, response additivity, Bliss independence) [52]. Among these, we chose the Bliss independence model [56], because for the tested therapies we assume two independent pathways with sigmoidal dose-responses and the same maximum effect (100% removal of Aβ). Therefore, we designed an experiment with doses for the single agents which should result in an approximately 30% pGlu3-Aβ reduction. In the case of Bliss additivity (CI_Bliss_ = 1), the combination of the same doses of both agents should result in an ≈51% reduction of pGlu-Aβ (see Equation (1)) whereas lower or higher values for the CI_Bliss_ are indicative for synergism or antagonism, respectively.

We used this model here to evaluate a combination study of an antibody targeting pGlu3-Aβ, m6 (murine PBD-C06), and Varoglutamstat (PQ912), the first in class glutaminyl cyclase inhibitor in clinical development for treatment of AD [32]. Thereby, the aims of the study are two-fold: (a) To test the hypothesis that pGlu-Aβ antibodies and PQ912 address independent pathways in reducing pGlu3-Aβ pathology and (b) if so, to show that the combination of low, borderline effective doses of the single agents PQ912 or m6 antibody exert a significant effect on pGlu-Aβ in combination. To address these questions, we used a transgenic model with enhanced pGlu3-Aβ formation, the hAPPsl×hQC double transgenic mouse [27]. The treatment started at an age of 9 months, where the mice already showed amyloid pathology but still accumulate Aβ over the 16-week treatment period, thus mirroring progress of Aβ pathology similar to very early stages (preclinical) of human AD. For either of the single treatments at the applied doses, the analyses of the brain tissue by ELISA and histology consistently revealed an insignificant decrease of pGlu3-Aβ and total Aβ, whereas the combination of both agents at the same doses generally shows a stronger response than the best single agent treatment. The response exerted by the combination treatment becomes significant in the majority of analyses. According to Foucquier and Guedj [52] these results fulfill the requirements for additivity for both the “highest single agent” as well as the “combination sub-thresholding” approach. The assessment of the Bliss combination index revealed values of nearly 1, which are indicative for an additive effect of the test compounds addressing two independent pathways both targeting pGlu-Aβ pathology (Table 1). Furthermore, the use of two independent techniques (ELISA, histochemistry) for evaluation of the efficacy of the treatments on reduction of Aβ pathology enables a robust conclusion.

Due to the limited number of animals enrolled in the treatment, the study has some limitations which may be addressed in future studies. The number of dose levels for the single and combination treatments should be increased to identify the most effective dose combination. This also allows the use of a dose-effect based strategy according to Chou [37] for evaluation and could directly demonstrate that a combination with half of the doses of the single treatments will result in same effect as the single treatments. Furthermore, it should be demonstrated also for the combination that the reduction of Aβ leads to an adequate improvement of memory function.

Nevertheless, the data strongly support an additive effect of anti-pGlu-Aβ antibody treatment and glutaminyl cyclase inhibition, i.e., both molecular treatment paradigms basically address different and independent mechanisms to suppress pathology-related pGlu-Aβ and general Aβ accumulation. This conclusion can be explained by the processes, in which both compounds act: QC inhibition interferes with de novo synthesis of pGlu-Aβ, the antibody requires the prior formation of the molecule for binding and clearance by microglia or eventually invading monocytes (Figure 3). Although this conclusion appears straightforward, it should be considered that QC inhibition has been shown to also exert anti-inflammatory effects, mainly driven by suppression of CCL2, resulting in inhibitory effects on monocyte migration and activation [57,58]. Although this will be covered in the upcoming studies, the concomitant suppression of inflammation by QC inhibitors without negatively effecting amyloid clearance by phagocytosis might be also favorable for combination with passive immunotherapy to potentially avoid adverse effects such as amyloid-related imaging abnormalities (ARIA) as seen in most clinical trials with plaque-binding anti-amyloid antibodies.

Considering that a part of pGlu-Aβ is formed during the secretion process, the combination therapy of these two paradigms might prove favorable to efficiently suppress the accumulation of this toxic species, since it is anticipated that the antibodies are ineffective in clearing intracellular amyloid peptides, as anticipated for tau immunotherapy [32]. The small molecule inhibitor has been shown to penetrate cellular barriers efficiently. Hence, we propose two potential outcomes of a combination of Varoglutamstat with monoclonal antibodies: (i) Reduction of the dose of each compound while maintaining the therapeutic effect by combination treatment as a matter of additivity; and, (ii) possibly, use of the small molecule QC inhibitor after antibody or combination treatment for long-term prevention of pGlu3-Aβ toxicity (“treat and maintain”). Although requiring further analysis and additional validation in clinics, we here provide a strong rationale for the evaluation of additional drug combinations for treatment of AD.

## 4. Materials and Methods

### 4.1. Materials

The glutaminyl cyclase inhibitor PQ912 [(*S*)-1-(1H-benzo[d]imidazol-5-yl)5(4-propoxyphenyl)imidazolidin-2-one] was used in this study. PQ912 was produced by Carbogen Amcis (Aarau, Switzerland). Chow containing 0.8 g PQ912 per kg was prepared by Ssniff Spezialdiäten (Soest, Germany) on base of the Ssniff R/M-H (V1534) diet.

The mouse IgG2a variant of the murine anti-pGlu3Aβ antibody PBD-C06 (PBD-C06.02; m6) has been produced as described previously [34,59].

### 4.2. Animal Experiments and Tissue Collection

hAPPsl×hQC mice were used in this study [27]. These mice express human amyloid precursor protein (hAPP)695 bearing the Swedish and London (sl) mutation and human QC, both under control of the Thy-1 promoter. The mice develop pGlu-Aβ positive deposits starting at an age of 6 to 7 months, the percentage of pGlu3-Aβ ranges, depending on the age of the animals, from 0.1–2% (https://qpsneuro.com/in-vivo-services/animal-models/app-transgenic-mouse-models/appsl-x-hqc-transgenic-mouse-model/, access on 28 October 2021). Animals were housed at QPS Austria (Grambach) in individually ventilated cages on standardized rodent bedding supplied by J. Rettenmaier & Söhne (Rosenberg, Germany) (21 °C, 40–70% rel. humidity, lights on from 6 a.m. to 6 p.m.). Pelleted standard rodent chow or compound containing chow (Ssniff, Soest, Germany) as well as normal tap water was available ad libitum.

In an initial dose-finding study, eight hAPPsl×hQC mice per group (male and female) were used. Animals were treated for 16 weeks starting at an age of 9 ± 1 months. The three groups were either treated with vehicle, 150 µg anti-pGlu3-Aβ antibody (m6), or 500 µg m6 per animal once a week by i.p. injection (application volume 200–250 µL). At the end of the study, animals were sacrificed, brains collected and subjected to Aβ extraction followed by ELISA analysis as described below [27].

In a combination study investigating the efficacy of QC inhibitor (PQ912) and anti-pGlu3-Aβ antibody (m6), 70 hAPPsl×hQC mice of both sexes aged between 8 and 9 months were allocated to five treatment groups (Table 2). Either IgG2a isotype control (C, E) or m6 (D, F) or vehicle (B) was administered once a week i.p. for 16 weeks. In addition to the i.p. treatment, two groups of animals (E and F) received PQ912 orally via food pellets for the 16-week treatment period. Body weight and food consumption was assessed once a week. Another subset of 10 naïve animals was subjected to tissue sampling at the time point of treatment start (Baseline group A). Because of premature death, one additional animal was included in group F to compensate for the lost animal.

At the end of the 16-week treatment period, animals were sacrificed and plasma and brain tissue collected and analyzed as described previously [27].

### 4.3. Aβ Extraction and ELISAs

Briefly, for Aβ extraction, one brain hemisphere without cerebellum was homogenized in TBS (20 mM Tris, 137 mM NaCl, pH 7.6; 2.5 mL, Dounce homogenizer) containing protease inhibitor cocktail (Complete Mini, Roche) and 0.1 mM AEBSF, sonicated and centrifuged at 75,500× *g* for 1 h at 4 °C. The supernatant (TBS soluble Aβ fraction) was stored at −80 °C. Insoluble Aβ species were sequentially extracted from the pellet using 2.5 mL 2% SDS in distilled water (SDS fraction), and 0.5 mL 70% formic acid (FA fraction). The formic acid extract was neutralized by addition of 3.5 M Tris solution and diluted to a final volume of 12.95 mL using ELISA blocking buffer (Pierce, Cat.No. 37571, ThermoFisher, Dreieich, Germany). Aβ(x–42) and pGlu3-Aβ(3–42) were determined in all brain fractions by specific sandwich ELISAs (IBL, Hamburg, Germany) according to the manufacturer’s manual. Individual brain content of Aβ (ng/g) was calculated by considering the brain wet weight of each sample. The sum of Aβ detected in the SDS and FA fractions were considered as the insoluble pool of Aβ.

### 4.4. Immunohistochemistry

Immunohistochemistry using the antibodies K17 (pGlu3-Aβ specific, IgG2b, Vivoryon N.V., Germany) and 82E1 (Aβ(1–x) specific; IBL Hamburg, Germany) was performed as described previously [35].

Briefly, 10-μm-thick brain sagittal cryosections were cut using a Leica CM1850 cryostat and mounted on Colorfrost Plus slides (Fisher Scientific). Sections were air-dried for 20 min and washed with TBS for 2 × 5 min. Immunohistochemistry was carried out using the Elite ABC Kit (Vector Laboratories, Burlingame, CA, USA). Quantitative image analysis of the percent area of immunoreactivity (IR) was obtained by using the Bioquant image analysis software (Nashville, TN, USA) on six sections per mouse at three equidistant planes for hippocampus and frontal cortex.

### 4.5. Data Handling, Statistics, and Calculations

The data obtained from the initial dose finding study were analyzed using ANOVA and Dunnett’s post-hoc comparison of m6-treated groups with the vehicle-treated group.

For statistical data evaluation of efficacy in the combination study, a stepwise approach was used. First, a 2-way-ANOVA with the baseline, the PBS-treated, and the isotype-antibody-treated group using time and gender as factors was done to evaluate whether the animals develop Aβ pathology over duration of treatment and if there were gender differences or differences between PBS and isotype treatment. This analysis was performed for the insoluble Aβ fractions. A significant (α = 0.05) increase of pathology after 16 weeks compared to baseline, as observed for the female mice, was considered as prerequisite for further analysis. In the second step the groups treated with a single agent (m6 or PQ912) and the combination treatment were compared with the isotype antibody treated control using one-way ANOVA and Dunnett’s post-hoc comparison of treatment groups with the isotype control group (respective non-parametric tests for histochemical analyses). Very low total Aβ42 levels were detected at end of treatment in three 12 months old animals (one male and one female, group E and one female, group D). In these mice the total Aβ42 levels were lower than half of the lowest value in the baseline group (433 ng/g). It is assumed that these animals did not express the APP transgene sufficiently. Therefore, these animals were excluded from further analyses.

For statistical evaluations and preparation of graphs Prism 7 for Windows (V7.03, GraphPad Software Inc., San Diego, CA, USA) was used.

#### Evaluation of Additivity

Additivity of treatments was evaluated according to the Bliss model which assumes that both treatments independently contribute to a common result [52]. The Bliss combination index (CI_Bliss_) was calculated according to the following formula:CI_Bliss_ = (E_A_ + E_B_ − E_A_ ∗ E_B_)/E_AB_(1)
were E_A_, E_B_, and E_AB_ are the effects of treatment A, treatment B, and the combination treatment AB, respectively. Using this equation, a combination index of 1 indicates additivity of the single treatments in the combination, a CI_Bliss_ < 1 indicates synergism and an index > 1 antagonism. The effect was calculated as % decrease in Aβ compared to the isotype control:E (%) = 100 × (1 − Aβ_Treatment_/Aβ_Isotype control_)(2)

## Figures and Tables

**Figure 1 ijms-22-11791-f001:**
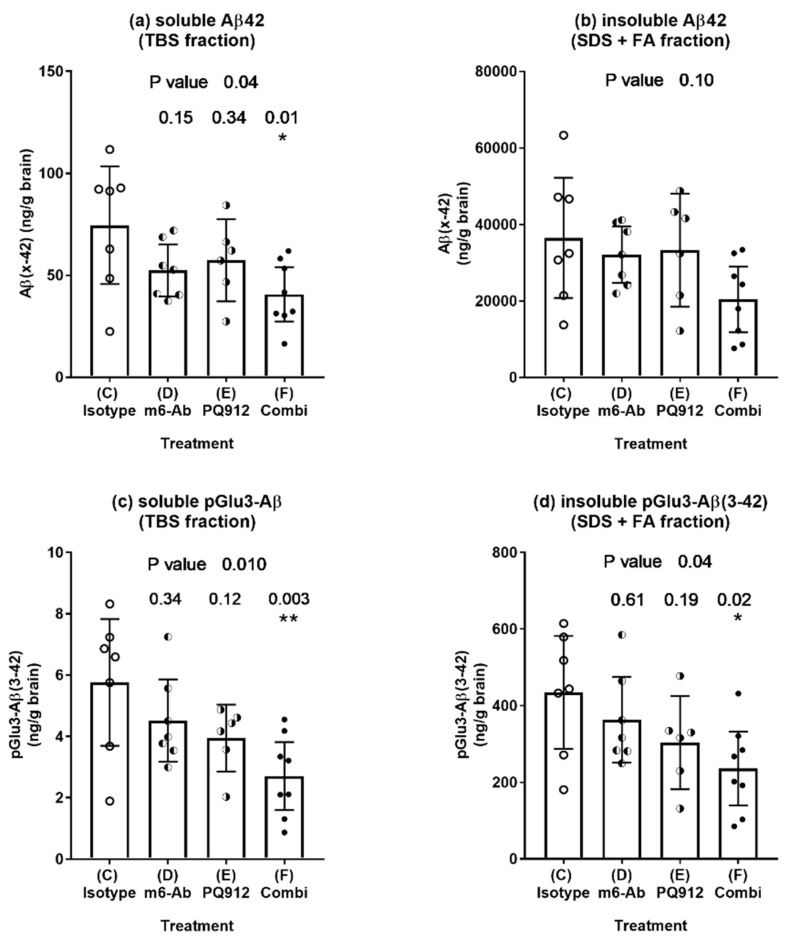
Aβ levels (ELISA) in brain homogenates of hAPPsl×hQC mice after treatment with PQ912 (Varoglutamstat), a monoclonal pGlu-Aβ antibody (m6-Ab) or in combination for 16 weeks. Concentration of Aβ(x–42) (**a**,**b**) and pGlu3-Aβ(3–42) (**c**,**d**) in soluble (**a**,**c**) and insoluble (**b**,**d**) brain fractions of 12-months-old hAPPsl×hQC mice after 16 weeks of treatment. Dots represent individual levels. Bars and whiskers represent mean ±95% confidence interval (CI). ANOVA *p*-values are given on top of the graphs. Numbers above the bars represent adjusted *p*-values of Dunnett’s post-hoc comparison with the isotype control (*—*p* < 0.05, **—*p* < 0.01). There is a nominal decrease of total Aβ42 and pGlu3-Aβ42 by the single treatments. The effect of the combination is stronger compared to the single treatment and becomes significant for pGlu3-Aβ in both fractions and for Aβ42 in the TBS fraction.

**Figure 2 ijms-22-11791-f002:**
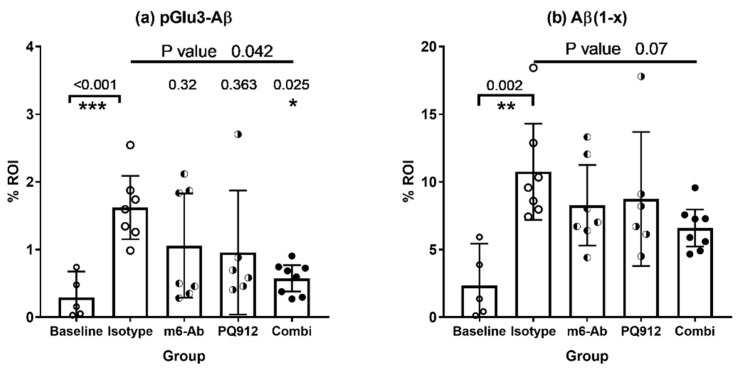
Quantitative immunohistochemistry of brains from hAPPsl×hQC mice after treatment with PQ912 (Varoglutamstat), a monoclonal pGlu-Aβ antibody (m6-Ab) or in combination for 16 weeks. (**a**) K17 stains pGlu3-Aβ, (**b**) 82E1 stains the Aβ N-terminus starting with amino acid 1 [Aβ(1–x)]. Significant differences between isotype control and baseline (*t*-test) indicate a development of Aβ pathology over the treatment period. In comparison to the isotype control, nominal lower mean Aβ staining was found in case of all treatments. The combination treatment resulted in significantly reduced staining for pGlu3-Aβ. Kruskal–Wallis test for group comparison (top line *p*-value) with Dunn’s test for post-hoc comparison with isotype control (*p*-values above bars) (*—*p* < 0.05, **—*p* < 0.01, ***—*p* < 0.001).

**Figure 3 ijms-22-11791-f003:**
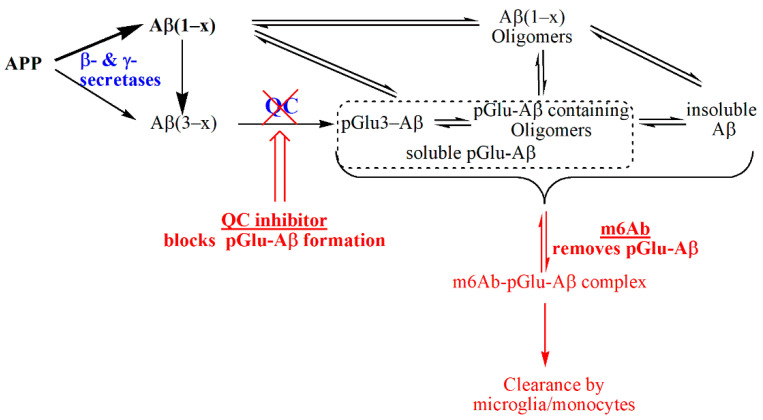
Schematic representation of the molecular pathways for reduction of pGlu3-Aβ by QC inhibition and anti-pGlu3-immunotherapy (m6Ab). The current study supports the concept that both strategies act independently and are additive. The formation of pGlu-Aβ occurs after APP cleavage by sheddases such as BACE1 or meprin β, which may lead directly to N-truncated Aβ forms and the pGlu3-Aβ precursor, Aβ(3–x). While QC inhibition prevents de novo-synthesis of pGlu3-Aβ, which reduces formation of toxic oligomers and coaggregation with other Aβ forms, the pGlu3-Aβ specific antibody prevents aggregation and elicits clearance of extracellular, soluble pGlu3-Aβ containing aggregates by opsonization and phagocytosis by microglia and/or monocytes.

**Table 1 ijms-22-11791-t001:** Quantitative evaluation of the combination treatment with PQ912 and m6-Ab for 16 weeks, applying the Bliss model. The parameters are: % mean response (E_x_ = % decrease of Aβ in respective group x compared to isotype control) for different groups, and calculated Bliss additivity and Bliss combination index (CI_Bliss_) for Aβ42 and pGlu-Aβ42 in soluble and insoluble brain fractions; CI_Bliss_ are given, if a robust (significant ANOVA) effect on Aβ level was observed (compare to Figure 1 and Figure 2).

	m6-Ab (E_D_)	PQ912 (E_E_)	Combi (E_F_)	Bliss Additivity (E_D_ + E_E_ − E_D_ ∗ E_E_)	CI_Bliss_ (E_D_ + E_E_ − E_D_ ∗ E_E_)/E_F_	Rating of CI According to Chou [37]
Aβ biochemistry						
soluble pE-Aβ42	21.6	31.5	53.0	46.3	0.87	slight synergism
insoluble pE-Aβ42	16.4	30.2	45.7	41.6	0.91	nearly additive
soluble Aβ42	29.8	23.1	45.4	46.0	0.99	(nearly) additive
Aβ Histochemistry						
pE3Aβ (K17)	34.6	41.0	64.5	61.4	0.95	nearly additive

**Table 2 ijms-22-11791-t002:** Overview on treatment groups in the combination experiment.

Group	N	Treatment		Time of Tissue Sampling
		Chow	Injection (i.p.)	
A	10	-	-	baseline
B	10	vehicle	PBS	at the end of treatment
C	15	vehicle	IgG2a isotype control (150 µg/week)	at the end of treatment
D	15	vehicle	m6 (150 µg/week)	at the end of treatment
E	15	PQ912 (0.8 g/kg pellets)	IgG2a isotype control (150 µg/week)	at the end of treatment
F	15 + 1	PQ912 (0.8 g/kg pellets)	m6 (150 µg/week)	at the end of treatment

## Data Availability

Data of the study are available from the corresponding authors upon reasonable request.

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
