# Peer review of "Combination of the Glutaminyl Cyclase Inhibitor PQ912 (Varoglutamstat) and the Murine Monoclonal Antibody PBD-C06 (m6) Shows Additive Effects on Brain Aβ Pathology in Transgenic Mice"

_ijms, 2021, doi:10.3390/ijms222111791_

Round 1

Reviewer 1 Report

The authors have carried out an accurate work analyzing the therapeutic potential of 2 compounds: Glutaminyl Cyclase Inhibitor PQ912 (Varoglutamstat) and the Murine Monoclonal Antibody PBD-C06 (m6) which aim at reducing the levels of pyroglutamate-modified amyloid peptides.

It is a simple and concise work with what it is intended to demonstrate. This reviewer only has a number of questions and suggestions.

  • The abstract provides little or no concrete information about the results, I suggest a rewrite where the main results of the study are included in a concrete way that allows to have a previous idea before reading the article.
  • The paper describes the use of 2 treatments, but it is striking that the administration to the animals is done differently depending on the treatment, which could be causing certain interferences due to a different uptake of the treatment. Why is it that PQ912 is administered in the feed and the M6 antibody i.p.? 
  • Please include in the methodology how you quantified the positive brain area in the immunohistochemistry, number of slices per animal and type of quantification.

  • The work describes a significant reduction in pGlu-Aβ levels with respect to the control, but is this reduction sufficient to produce a therapeutic effect in the animals at the functional level? The work should be accompanied by a functional evaluation of the animals and establish the true impact of pGlu-Aβ reduction on disease progression.

Minor corrections

Introduction and Results section correct the reference error "(Error! Reference source not found.)"

Author Response

Dear Sir/Madam,

we thank you for your time and valuable comments to improve our manuscript. Please find attached the statements to your concerns in a point by point fashion

  1. Schilling

Reviewer 1:

The authors have carried out an accurate work analyzing the therapeutic potential of 2 compounds: Glutaminyl Cyclase Inhibitor PQ912 (Varoglutamstat) and the Murine Monoclonal Antibody PBD-C06 (m6) which aim at reducing the levels of pyroglutamate-modified amyloid peptides.

It is a simple and concise work with what it is intended to demonstrate. This reviewer only has a number of questions and suggestions.

1. The abstract provides little or no concrete information about the results, I suggest a rewrite where the main results of the study are included in a concrete way that allows to have a previous idea before reading the article.

Answer: The abstract was revised, now including the major results.

2. The paper describes the use of 2 treatments, but it is striking that the administration to the animals is done differently depending on the treatment, which could be causing certain interferences due to a different uptake of the treatment. Why is it that PQ912 is administered in the feed and the M 6 antibody i.p.? 

Answer: The different forms of application are simply explained by the different nature of the used molecular entities, one a small molecule and the other a biologic, taking their respective pharmacokinetic feature into consideration.  It has been shown that the small molecule, which has a short half live in mice, shows the most stable exposure when applied by chow. The level of PQ912 achieved with certain doses have been documented in an earlier paper (Hoffmann et al. 2017). There is of course a need to apply the biologic parenteral. The adequate vehicle chow without PQ912 was used for antibody single and vehicle treatment group and vehicle ip injection was used for the single and vehicle treatment group whereas the PQ912 single treatment and the vehicle group received an isotype antibody injection with identical formulation as the active m6 antibody.

3. Please include in the methodology how you quantified the positive brain area in the immunohistochemistry, number of slices per animal and type of quantification.

Answer: The immunohistochemistry and quantification methods was updated and described more in detail.

4. The work describes a significant reduction in pGlu-Aβ levels with respect to the control, but is this reduction sufficient to produce a therapeutic effect in the animals at the functional level? The work should be accompanied by a functional evaluation of the animals and establish the true impact of pGlu-Aβ reduction on disease progression.

Answer: Behavior was not assessed in this study and the study was not powered for behavioral investigations. This study was primarily set up to demonstrate the effect of PQ912 (Varoglutamstat) and its combination with an antibody on Aβ accumulation in the brain in order to collect evidence, whether these treatments can act in an additive or synergistic manner. Because this conclusion can be drawn by biochemical assessments, we did not increase the number of animals. However, it should be noted that in former experiments it could be demonstrated in different AD animal models [Hoffmann et al. 2017, Crehan et al. 2020], that the treatment with the single agents PQ912 or m6 are capable to reduce pEAβ and concomitantly improve behavior in the Morris-water-maze. Therefore, we generally expect such a positive therapeutic effect in response to the Aβ reduction, which is now worth to be investigated for the combination more in depth. In order to address the criticism of the referee, we added a paragraph citing the previous studies including the behavioral analysis.

Minor corrections

  • Introduction and Results section correct the reference error "(Error! Reference source not found.)"

Answer: Thank you very much, we apologize for this. The error was found only in the result section and corrected.

Reviewer 2 Report

In this article the authors demonstrate that the combination of the glutaminyl cyclase inhibitor PQ912 (Varoglutamstat) and the murine monoclonal antibody PBD-C06 (m6) shows additive effects on reducing Aβ pathology in a transgenic mouse model of Alzheimer's disease (AD). They propose that PQ912 prevents the formation of pGlu3-Aβ while PBD-C06 clears existing pGlu3-Aβ deposits by eliciting phagocytosis by microglia/monocytes. 

This finding is interesting as it supports the potential use of this combination to reduce amyloid pathology in AD. However, as the authors have suggested similar combinations aiming to reduce amyloid pathology are already in clinical trials for AD so the findings stated here are expected.

In order to improve the article, I have several recommendations:

   1. I consider that it would be essential to check if the significant reduction in amyloid pathology observed with the combination PQ912 and PBD-C06 is able to restore or at least improve memory function.

   2. You mentioned that the mode of action of both treatments is different. One compound reduce the production of pGlu3-Aβ while the other clears existing pGlu3-Aβ deposits. However, you did not find any difference between both treatments neither in the ELISA or by IHQ. Is this just because the doses used are sub-therapeutic? Or it would be necessary to do further experiments, such as analyse specifically the number of senile plaques to see differences? It would be also interesting to do double IF assays with antibodies against microglia and pGlu3-Aβ to demonstrate that m6 antibody increases microglial Aβ engulfment.

3. In the introduction, I would suggest to include a brief sentence explaining which are the main pathological hallmarks or AD, instead of starting directly by mentioning the results of the clinical trials. Also, I would explain what Lecanemab or Donanemab do.

4. I think that it would be nice to subdivide the results into different sub-sections to facilitate the reading.

5. It would be great to include the statistical results in the text (this was only done in some results, not in all significant ones).

6. I believe that figures look better if you include bars with the correspondent asterisks to indicate significant differences.

7. In the case of the IHQ assays, it would be great to see representative images of each group.

8. In the methods, it would be great to have the ELISA and IHC assays explained in separate sections. Also, even if you refer to another article for the methods it is nice to give a small sum up of the technique used.

Author Response

Dear Sir/Madam,

we thank you for your time and valuable comments to improve our manuscript. Please find attached the statements to your concerns in a point by point fashion.

On behalf of all authors, sincerely,

S. Schilling

In this article the authors demonstrate that the combination of the glutaminyl cyclase inhibitor PQ912 (Varoglutamstat) and the murine monoclonal antibody PBD-C06 (m6) shows additive effects on reducing Aβ pathology in a transgenic mouse model of Alzheimer's disease (AD). They propose that PQ912 prevents the formation of pGlu3-Aβ while PBD-C06 clears existing pGlu3-Aβ deposits by eliciting phagocytosis by microglia/monocytes. 

This finding is interesting as it supports the potential use of this combination to reduce amyloid pathology in AD. However, as the authors have suggested similar combinations aiming to reduce amyloid pathology are already in clinical trials for AD so the findings stated here are expected.

Comment: Dear Sir/Madam, we highly appreciate your statement but would like respond on this point.  The statement of similar combinations is true at a general level related to potential advantages of a combination addressing independent pathways of the Abeta pathology. To our knowledge, however, there are no data of a combination therapy yet available verifying the hypothesis of approaches interfering with pE-Abeta. The two treatments (QC inhibition and pGlu-immunotherapy) are new to the field and entered clinical testing not long ago. The results here are also not straightforward to conclude, because other combinations newer addressed the same amyloid species. Insofar, we feel the results could not have been expected and are very important for future combination treatments.  

In order to improve the article, I have several recommendations:

1. I consider that it would be essential to check if the significant reduction in amyloid pathology observed with the combination PQ912 and PBD-C06 is able to restore or at least improve memory function.

Answer: A very similar concern was raised by reviewer 1. We agree with this recommendation, but as mentioned above the current study was designed to investigate only the primary pharmacological effect on Aβ deposition. We showed already with each single compound in more than one study that reduction in pE-Abeta goes along with an improvement in memory function in mice. We think it is reasonable to expect that a similar reduction in pE-Abeta by a combination would lead to comparable memory benefits. Nevertheless, to verify the effectiveness of the combination could be now aim of future studies. A paragraph addressing the limitations of the study was added to the discussion.

2. You mentioned that the mode of action of both treatments is different. One compound reduce the production of pGlu3-Aβ while the other clears existing pGlu3-Aβ deposits. However, you did not find any difference between both treatments neither in the ELISA or by IHQ. Is this just because the doses used are sub-therapeutic? Or it would be necessary to do further experiments, such as analyse specifically the number of senile plaques to see differences? It would be also interesting to do double IF assays with antibodies against microglia and pGlu3-Aβ to demonstrate that m6 antibody increases microglial Aβ engulfment.

Answer: We fully agree to the observation and conclusion of the reviewer. The doses applied with both single treatments were adjusted by purpose on sub-therapeutic level. The dose selection was based on previous examinations (Hoffmann et al., 2017 for Varoglutamstat; Crehan et al. 2020 for m6). The selection is subtherapeutic to reveal the therapeutic advantage and the type of combination properly. Higher doses would lead to a “ceiling effect” preventing a reliable conclusion on the mode of combination. With the current setup, doses and parameters analyzed, we get the similar net effect of the single treatments on Aβ deposition. We included a statement in the results section. With regard to the immunohistochemical staining, we would make this reviewer aware of our numerous previous publications on pGlu-immunotherapy. The microglial engulfment was investigated, for instance, in Crehan et al. 2020.

3. In the introduction, I would suggest to include a brief sentence explaining which are the main pathological hallmarks or AD, instead of starting directly by mentioning the results of the clinical trials. Also, I would explain what Lecanemab or Donanemab do.

Answer:  We are grateful for this suggestion and included an introductory paragraph in the introduction.

4. I think that it would be nice to subdivide the results into different sub-sections to facilitate the reading.

Answer:  We divided the manuscript into subsections according to the suggestion.

5. It would be great to include the statistical results in the text (this was only done in some results, not in all significant ones).

Answer:  The statistical analysis is now stated in the text consistently.

6. I believe that figures look better if you include bars with the correspondent asterisks to indicate significant differences.

Answer:  The figures have been revised according to the suggestion of the reviewer. 

7. In the case of the IHQ assays, it would be great to see representative images of each group.

Answer:  We included these images into a new supplementary figure. 

8. In the methods, it would be great to have the ELISA and IHC assays explained in separate sections. Also, even if you refer to another article for the methods it is nice to give a small sum up of the technique used.

Answer:  A similar concern was raised by reviewer 1, we separated the description and added details as requested.